# Patients' initial steps to cancer diagnosis in Denmark, England and Sweden: what can a qualitative, cross-country comparison of narrative interviews tell us about potentially modifiable factors?

John MacArtney,[1] Marlene Malmström,[2,3] Trine Overgaard Nielsen,[4] Julie Evans,[1] Britt-Marie Bernhardson,[5] Senada Hajdarevic,[6] Alison Chapple,[1] Lars E Eriksson,[5,7,8] Louise Locock,[9] Birgit Rasmussen,[2,10] Peter Vedsted,[4,11] Carol Tishelman,[5,12] Rikke Sand Andersen,[4] Sue Ziebland[1]

For numbered affiliations see end of article.

**Correspondence to**
Dr John MacArtney;
john.macartney@phc.ox.ac.uk

## ABSTRACT

**Objectives** To illuminate patterns observed in International Cancer Benchmarking Programme studies by extending understanding of the various influences on presentation and referral with cancer symptoms.

**Design** Cross-country comparison of Denmark, England and Sweden with qualitative analysis of in-depth interview accounts of the prediagnostic process in lung or bowel cancer.

**Participants** 155 women and men, aged between 35 and 86 years old, diagnosed with lung or bowel cancer in 6 months before interview.

**Setting** Participants recruited through primary and secondary care, social media and word of mouth. Interviews collected by social scientists or nurse researchers during 2015, mainly in participants' homes.

**Results** Participants reported difficulties in interpreting diffuse bodily sensations and symptoms and deciding when to consult. There were examples of swift referrals by primary care professionals in all three countries. In all countries, participants described difficulty deciding if and when to consult, highlighting concerns about access to general practitioner appointments and overstretched primary care services, although this appears less prominent in the Swedish data. It was not unusual for there to be more than one consultation before referral and we noted two distinct patterns of repeated consultation: (1) situations where the participant left the primary care consultation with a plan of action about what should happen next; (2) participants were unclear about under which conditions to return to the doctors. This second pattern sometimes extended over many weeks during which patients described uncertainty, and sometimes frustration, about if and when they should return and whether there were any other feasible investigations. The latter pattern appeared more evident in the interviews in England and Denmark than Sweden.

**Conclusion** We suggest that if clear action plans, as part of safety netting, were routinely used in primary care consultations then uncertainty, false reassurance and the inefficiency and distress of multiple consultations could be reduced.

### Strengths and limitations of this study

► This study provides a social science informed, qualitative cross-country comparison of 155 indepth interviews with bowel and patients with lung cancer recruited within 6 months of a diagnosis in Sweden, Denmark and England.
► The methods we use provide insight into why and how potentially modifiable factors identified by the International Cancer Benchmarking Project—including response to symptom experiences, differences in willingness to consult, how people negotiate access to healthcare and what happens during consultations—affect the time to diagnosis in a primary care setting.
► The study was limited to patient interviews and no 'first-hand' observational data were available to compare accounts.

## INTRODUCTION
### Background
No single factor is likely to explain why differences in cancer outcomes persist between high-income countries with universal health coverage. The International Cancer Benchmarking Partnership (ICBP), which compares cancer registry data and cross-country surveys, has explored several hypotheses about why variations occur. These studies have shown a number of potentially modifiable factors, for example, that patterns in public knowledge about cancer awareness and beliefs were not clearly associated with variations in survival across countries.[1] The ICBP studies have also drawn attention to different patterns in public willingness to consult a Primary Care professional (PCP), with patients in the UK reporting particular concern 'not to waste the

doctor's time'.[1] There are also cross-country differences in general practitioners (GPs) who expressed willingness to refer a patient with suspicious symptoms at their first presentation, and PCPs have reported that investigations such as CT and MRI were harder to access, and results took longer in England and Denmark than in Sweden.[2] There seems to be a relationship between smoking behaviour and willingness to consult: in separate, recent studies, both English[3] and Danish smokers[4] have been shown to be less likely than non-smokers to consult with red flag lung cancer symptoms. It has been suggested that this is related to the shame, blame and stigma associated with lung cancer as a smoking-related disease.[5] Socioeconomic deprivation, inequalities in service provision and regional differences in expectations about the likely consequences of seeking care further contribute to a complicated picture.

Denmark, England and Sweden (all ICBP participants) were selected for this comparison because survival for lung and bowel cancer between 1995 and 2007 were persistently higher in Sweden than in Denmark or England, particularly in the first year after diagnosis.[6] We chose lung and bowel cancer because they affect both genders, are the two most common causes of death from cancer across Europe,[7] their symptomatology is often diffuse or vague and they are often prone to late-stage diagnosis. Earlier diagnosis has become a key healthcare research and policy target in all three countries.[8–10] Analyses of potentially modifiable factors have drawn attention to between-country differences in stage at diagnosis and access to good care, diagnostics and screening.[11] Late presentation and long diagnostic intervals are also potentially modifiable contributory factors to poor survival.[6 11–15] Cross-country comparative research can suggest routes for service redesign as well as hypotheses for further research.[16] Social and health sciences have suggested potentially modifiable factors affecting the time to diagnosis include public response to symptom experiences,[1 3–5 17–26] differences in willingness to consult,[1 3–5 17–24] how people negotiate access to healthcare,[1 18 22 27 28] what happens during consultations[2 27 29–31] and access to diagnostic tests.[2] While surveys and cancer registry data can provide excellent high-level comparative data showing patterns of association, in-depth qualitative research is needed to help illuminate why and how observed variations may occur.[32 33] We therefore designed a study to contribute a qualitative cross-country comparative analysis of narrative interviews with patients recently diagnosed with lung or bowel cancer in Denmark, England and Sweden. In this paper, the research question we ask is, what might explain some of the variations identified in the ICBP?

### The healthcare systems in Denmark, England and Sweden

All three countries have primarily tax-based health systems. However, in Sweden, patient copayments vary from no payment to Kr300 per primary care visit, depending on county council, up to a total of ~Kr1100 (£100) per year.[34] No copayments are made in England and Denmark, although there are prescription charges in all three countries. In Sweden two-thirds of GPs are publicly employed, whereas in Denmark and England most are self-employed.[34] Sweden spends over 20% more on healthcare per head of population than England.[35] Contact with primary care varies between the three countries: Swedish patients consult PCPs least frequently (2.9 per annum) followed by Denmark (4.7 per annum) and UK (5.0 per annum, which includes telephone consultations).[36] In 2015, England withdrew guidance stipulating access for urgent appointments within 48 hours.[34] In some settings in Sweden, a nurse is the first person a patient speaks to when seeking an appointment.[34] Most first contact in Denmark and England is with a receptionist, who may pass the patient to a nurse to be triaged. In Denmark, some receptionists have been trained to recognise potentially serious symptoms.[34] GPs have a central role in making referrals to lung and colorectal specialists in all three countries[34] The national cancer plan in Sweden is regionally administered and focused on secondary care,[37] whereas in England and Denmark, specific clinical guidance and targets for PCPs are also included.[34] England (implemented 2006) and Denmark (implemented 2014) have national cancer screening programmes for bowel cancer. In Sweden, bowel screening was implemented regionally, with some regions having some form of programme from 2008. Currently, it is recommended for those aged 60–74 years, although this again varies by region. Denmark has increased rapid access clinics and access to investigations, while in England, referral management systems have been interspersed between the GP and secondary care.[34 38]

## METHODS
### Sampling and recruitment

We interviewed 155 people during 2015 who were within 6 months of a diagnosis of lung or bowel cancer in the three countries. Purposive sampling[39] within each country was used to achieve gender balance and variation across age, urban and rural locations and pathway to diagnosis (see table 1). An experienced sociologist in England (JM) and an anthropologist in Denmark (TON) recruited and conducted all the interviews in their countries, while three experienced nurse researchers collaborated in different regions of Sweden (BMB, SH and MM). Participants were initially approached by treating clinicians in hospital clinics in all three countries. To reach data saturation, in England and Denmark, this approach was supplemented with some additional recruitment from support groups, social media and word of mouth. All potential participants were provided with an information sheet that included the rationale for the study and an explanation of what would be involved if they consented to take part in the study. Data saturation[40] was judged to have been reached in the analytical categories for all three countries before recruitment closed.

**Table 1** Participant demographic characteristics across the three countries.

| | | Denmark | | England | | Sweden | | Percentages | |
|---|---|---|---|---|---|---|---|---|---|
| | | BC | LC | BC | LC | BC | LC | BC | LC |
| Number of participants | | 28 | 22 | 25 | 20 | 30 | 30 | 54% | 46% |
| Female | | 13 | 8 | 12 | 10 | 14 | 15 | 47% | 46% |
| Age range | 31–50 | 2 | 0 | 4 | 2 | 2 | 2 | 10% | 5% |
| | 51–70 | 19 | 15 | 13 | 12 | 14 | 21 | 55% | 67% |
| | 71–90 | 7 | 7 | 8 | 6 | 14 | 7 | 35% | 28% |

BC, bowel cancer; LC, lung cancer.

## Data collection

Interviews took place in participants' homes, unless they preferred another location. One-to-one interviews were preferred, but a small number of participants requested a non-participating family member to be present. Interviews started with an open-ended question: 'Could you start by telling me, in your own words and in as much detail as you want, about everything that has happened since you first started to suspect there might be a problem with your health?' Interviews lasted between 45 min and 1.5 hours. During the interview, the researchers used a semistructured topic guide based on social science theories and the cancer research literature (highlighted in the Introduction), including factors related to the diagnostic interval.[1–5 17–31] The research team had extensive discussions about the topics to ensure comparable data were collected.[41] Interviews were audio recorded and transcribed verbatim.

## Analysis

Monthly teleconferences with the field research team (all of whom had a high level of spoken and written English) were held throughout the development of the topic guide, recruitment, data collection and analysis phases. Interview accounts were analysed for narrative themes that structured participant experiences. To do this, in each country, interview transcripts were imported into specialist computer software (NVivo V.10) for organising textual data and were coded by the interviewing researchers. The three research teams conducted separate thematic analysis with their own data using a coding frame developed through discussion in the teleconferences and based on the (anticipated) themes from the topic guide and on emergent themes.[42] One 2-day and one 4-day analyses and writing workshops were held with the research teams from all three countries. Anticipated issues drew on existing knowledge and theoretical insights.[1–5 17–31] Emerging analyses were iteratively tested within and between datasets. Allocations between the analysis categories (including identifying participants who described leaving the PCP consultation with an understanding of what should happen next and those who reported uncertainty about what to do) was discussed among the field researchers. Direct quotes from the interviews (translated into English by the bilingual researchers) are used to illustrate the results. Analysis and findings are presented in accordance with relevant Consolidated Criteria for Reporting Qualitative Research[43] criteria for reporting qualitative data and are consistent with interpretive approaches to analysis.[44]

## Patient involvement

Public and Patient Involvement (PPI) was conducted in accordance with good practice in each country. Representatives in England with experience of bowel or lung cancer were involved in preparing the funding application, the study design and the multidisciplinary advisory group, as well as advising on recruitment strategies, including the patient information sheets. In all three countries, PPI members were invited to comment on the draft interview topic guide. A group of three English PPI members met with the research team to look at a subset of the English data. Drafts of study papers have been circulated to the advisory panel. Summaries of the main study findings, prepared for a stakeholder final event in 2017, will be circulated to all study participants who expressed interest in seeing the results.

## Results

We compared the three sets of interviews to consider the following potentially modifiable steps, drawn from the literature, on the prediagnosis pathway: people's initial responses to bodily sensations and the transformation of these into symptoms justifying medical advice; accounts of accessing medical help; what happened in the consultation and planning for follow-up.

### Awareness and responses to signs and symptoms

The first potentially modifiable factors are the type of sensations and bodily changes people considered to justify seeking help. Participants in all three countries described considerable uncertainty about the possible implications of bodily sensations and when or if they should be considered symptoms warranting medical advice, especially if they felt well. For example, apart from some blood in his stools, a participant in England said he had been 'fit as anything and not feeling ill at all' (England—BC18). Even those who looked up their symptoms and risk factors online were reassured if they found that relatively few items on the checklists applied to them. People reasoned

that they must be low risk if they felt they had a good diet, were physically active, non-smokers (or long-term ex-smokers), generally fit, relatively young or without a family history of cancer. This reasoning suggested that some patients expected there to be a linear relationship between healthy input and outcomes. For example, a participant in Denmark who acknowledged "[I] smoke and [drink] alcohol and [eat] too much meat…" went on to suggest that other behaviours offered some protection from the profile of a typical cancer :

> Denmark—BC01: I have read about this, I'm very much into organic food. All the things you say can cause cancer… you don't know […]. I have at least lived a healthy life with vegetables everyday…and fruit, so maybe I'm not the typical cancer patient.

Persistent coughs, pains, stomach aches, changes in bowel habits, bleeding or constipation were attributed to everyday causes or to more familiar diseases such as influenza, pneumonia, irritable bowel syndrome (IBS) or haemorrhoids.[21] People with lung cancer, who smoked cigarettes, reasoned that a persistent cough was a 'normal' bodily response to smoking. Thus, even those who are aware that they have risk factors, or suspicious symptoms, may conclude that overall their risk is low and not seek medical help. Consistent with Zola's classic work on triggers to the consultation, when people did consult this was usually because the symptoms, interfered with daily life, became evident to others who encouraged them to consult or exceeded a self-imposed time limit[22] rather than they suspected that they might have cancer.

### Access to primary care

The second potentially modifiable factor is whether the participant felt able to access primary care. There were some apparent differences in accounts from the three countries. In England, people talked about long waiting times for 'non-urgent' appointments or (echoing ICBP findings) said they were reluctant to trouble busy GPs with potentially minor illnesses that might be treatable with over-the-counter remedies.[1] A participant in England, later diagnosed with bowel cancer, had concluded, 'It's haemorrhoids, leave it. Doctors are busy with other things' (England—BC20). Another participant in England with lung cancer explained why she had not consulted with a cough:

> England—LC12: When you hear that the GPs are so busy and being bothered by, you know, minor illnesses it makes you reluctant to go with something that you think [is minor] . . . I was, well, "What can the doctor do? They won't want to give you antibiotics for a virus." That was what I, I thought. And I just didn't want to really bother the doctor.

A participant in Denmark similarly reflected on an implication of GPs' time pressures:

> Denmark—LC15: The system is not always that easy. First you have to convince the secretary, that you need an appointment, right. That is what happens when they are too busy.

In contrast, we were struck that accounts in Sweden rarely included detailed justifications for using the doctor's time nor did they suggest that the onus was on the participant to determine whether the matter was serious or urgent. For example, a participant in Sweden diagnosed with bowel cancer said:

> Sweden—BC051: It began like this, that I found a little blood I can say in the stool, very little . . . And I tend to ignore such things but this was just strange enough, so I thought, "no, I've got to go to the health centre and hear what it is". And so I did.

Another participant in Sweden with lung cancer said, "It's better to go one time too many than one too few" (Sweden-LC102). This sentiment was not expressed in any of our interviews in England.

### During the consultation

The third potentially modifiable factor is what happens when the patient presents with symptoms. Seeking healthcare is not just a matter of making an appointment, but also of being able to present one's problem and be heard.[9] In all three countries, there were examples of a single consultation and a prompt referral. A participant in England said:

> England—BC18: Our local GP was brilliant, she said, "I'm not gonna do the kind of normal give you this stool sample packet, all that kinda stuff. I'm gonna hotline you straight to [hospital] down the road there for an endoscopy. Skip all the intermediate stuff and go straight into endoscopy," which was a fabulous decision, if you think of it.

Similarly a participant in Denmark explained,

> Denmark—BC13: The stomach pains had been going on for a while, but 1 day I just could not stand it anymore and drove straight down to my doctor's office. My doctor examined it and immediately said: "I will send you off to the hospital." And I was [sent off] in the afternoon. That was quick.

A participant in Sweden with bowel cancer recalled the GP saying, "No, I cannot see anything, but for safety's sake I will send you to a specialist […] so they get a closer look at it." (Sweden—BC051)

There were also examples of participants in all three countries who told us that they had returned to the doctors over periods of weeks or even months before they were referred for further investigations. Some participants in England and Denmark diagnosed with lung cancer said they had been repeatedly treated with antibiotics for chest infections. Participants diagnosed with bowel cancer from all three countries described

inconclusive consultations for presumed constipation, haemorrhoids, IBS, Crohn's or psychological causes; however, in our data, this appeared more often in interviews from Denmark and England. This example from England typifies this:

> England—BC09: . . . each time it was treated obviously as constipation. There was never even any mention of anything else. He's [the GP] a very, "I can do it. I can fix it. I can do it." But each time I come away, it was so sort of disheartening, because it seemed to be about what he could do and not actually helping the situation. But I did have, I had all sorts of laxatives from him. It's as if like we just keep trying to do this, but I was in agony each time I took it. I did explain that each time. . . But, yeah, nothing really. It was more like the repetitive sort of appointment.

### Making plans for follow-up

The fourth potentially modifiable factor is what happens after those consultations which do not include a referral. We looked in detail at the accounts of those participants who reported two or more presentations before referral and noted two different types of experience, depending on whether or not the patient reported leaving the consultation knowing what to do next.

### Repeated consultation(s) with patient awareness of next steps

First, there were participants in all three countries who described leaving the consultation with an understanding about what to expect, including how long to wait for symptoms to abate, what should prompt them to return and what would happen after that. For example, this participant in Sweden said:

> Sweden—LC129: And then the [GP] said, "No it's nothing. Your lungs sound very healthy, but you've had this cough for the last 8 weeks and if it does not disappear in another week, then please call us. Then we'll see."

The participants in this 'planned' group had also known what would be likely to happen if and when they did return; all three countries included examples of people who knew that further treatments or investigations would be needed if the symptoms did not resolve:

> England—LC17: And he [GP] prescribed 2 weeks antibiotics. And then he said that after that, if it didn't make any difference then he might prescribe omeprazole and maybe then a chest x-ray.

Follow-up advice can be reinforced by practice staff when delivering a normal test result in the course of their diagnostic investigations.[45] For example, a participant in Sweden reported that the practice nurse said, "if [it] gets worse—for example if you lose weight—contact us directly" (Sweden-BC055).

### Repeat consultation(s) with patient uncertainty about next steps

Second, there were participants who were unsure what would or should happen next, such as England—BC09 quoted above who described a 'disheartening' and 'repetitive sort of appointment'. These participants included those who worried that it was not appropriate to return to the doctor with the same symptoms and those who had been reassured at the initial consultation. The following participant in Denmark had requested a colonoscopy after many months of unusual symptoms and several investigations for other possible causes.

> Denmark—BC05: I can see now that I had these symptoms for 3 years. They [the doctors] thought that it was only the myalgia. But it probably was the colon cancer, even though I did not have the usual symptoms, you know. No bleeding, or diarrhoea or constipation. Just this strange heaviness in my lower parts. So I had an ultrasound of my bladder and kidneys, an x-ray of my lumbar area… And then I finally asked for a colonoscopy. My doctor asked me if I was sure… of course I was… but I had not thought of cancer.

In these accounts people had been unsure what to expect and left without a plan about what to do next. Unsurprisingly, people who had presented on two, three or more occasions described some frustration. In retrospect, some wondered if the PCP had decided that a 'test of time' or 'wait-and-see' was the best course of action, without sharing their reasoning.[30 31]

> England—BC03: And the simple thing and I agree the most obvious thing was, you know, piles and irritable bowel syndrome . . . I guess they, the thought was, "Well we'll send her away with that for now and if doesn't work, she's bound to come back."

Accounts of repeated consultations, without an action plan communicated to the patient, featured more strongly in the interviews with patients in England and Denmark than in Sweden.

Patients may welcome reassurance from their PCP about their suspicious symptoms, and this may not be a problem if the practitioner takes responsibility for following up the patient as part of safety netting—the process of communicating with patient what to expect, documenting any action plans and following the patient up as agreed.[46] A Swedish lung cancer patient explained that the GP had telephoned to see 'how she was' after a gastroscopy and made an urgent referral:

> Sweden—LC123: I got a referral to ultrasound of my thyroid and upper gastro. And then they discovered in a gastroscopy that I had oesophagitis … [a week later the GP] called to ask how I was. I said, "now it is bad," I said. "I think I have to go to the hospital urgently as it is now, it's … I cannot breathe." "Come here", she said, "and we'll make an urgent referral".

## DISCUSSION
### Principal findings
Our international qualitative comparison has considered four potentially modifiable features of the prediagnostic experience of lung and bowel cancer patients. Across the countries, there are many similarities in how symptoms were experienced and in awareness of what the symptoms might mean. Our analysis suggests differences between countries in people's accounts of their willingness to consult and reconsult, their descriptions of what happened during consultations and for those who did not receive an immediate referral, their confusion or clarity about what they should look for, when to return and what to expect. Reluctance and practical barriers to consulting the GP, and uncertainty about whether and when to reattend after an initial consultation, appeared more evident in interviews in Denmark and England. This indicates the benefit of communication of clear action plans to patients, including information on symptom development and other safety netting approaches,[47] which could well be developed more. PCPs action plans might include a 'test of time', yet this reasoning should be clearly communicated to the patient. This could also mean that patients would receive more timely referrals and avoid the repeated visits to primary care which featured particularly in English and Danish interviews.[31]

### Strengths and weaknesses of the study
The study design—a cross-cultural comparison using in-depth qualitative interview—is the first of its kind to study the route to cancer diagnosis. Our findings suggest potentially modifiable factors that could improve timely diagnosis of cancer in all three countries, although the scope for improvement may be greater in England and Denmark.

This qualitative study was designed to illuminate findings from the ICBP studies and is intended to be indicative rather than conclusive. Findings extend observations from other research that highlights the centrality of the primary care encounter in prediagnostic experiences. Participants' cancer disease stage is likely to have affected their experiences. To achieve diversity, we recruited people who were diagnosed with very early stage and others diagnosed with advanced disease in all three countries. Participants who were recruited via social media in Denmark and England may also be thought to have had different experiences than those recruited via clinics. All participants were, or had been, under specialist cancer care and within 6 months of diagnosis. Each data set was analysed by the country specific team, thus benefiting from familiarity with healthcare context, language and culture. These country-specific analyses were shared as part of a careful collaborative process, involving workshops, teleconferences and several drafts and further targeted analyses. Along with the teleconferences, these meetings helped to mitigate any potential for differences in the data arising from differences in country or disciplinary approaches to interviewing or analysis.[41]

### Strengths and weaknesses in relation to other studies
This study was designed to explore whether a cross-country comparison of qualitative interviews could help to illuminate findings from the ICBP survival analyses and surveys. While the high-level comparisons of large quantitative databases and surveys provide invaluable information, qualitative studies can suggest why these patterns may occur and identify avenues for service redesign, such as action plans, some of which could be implemented without further research. We did not observe the consultations described by the patients and it is possible that the PCPs involved did try to communicate an action plan, even if it did not feature in the patient's account. Again, we have no reason to suspect that variations in recall would differ between the countries. It is well known that public knowledge of cancer symptoms is insufficient to prompt people to seek help, as recently demonstrated in ICBP surveys.[1] Our work was guided by insights from the social and healthcare sciences into the complexity inherent in decisions to consult. These are influenced by people's ideas about their own candidacy, related to family history and health behaviours, as well as their access to care.[25 28 48] Evans et al have shown that patients, and their doctors, may be falsely reassured if a symptom comes and goes, reasoning that intermittent symptoms are unlikely to be serious.[24] Renzi and colleagues have drawn attention to the potential negative consequences of reassurance through an 'all-clear' result from investigations, which could be given with or without an encouragement to return.[45]

This study provides further insight into ICBP findings of how cancer symptoms are managed in primary care, in particular, how delays may be compounded when a patient is not referred promptly and no clear plan is communicated about the circumstances in which they should return for further consultation or investigations.[2 49] Current research is exploring how and why planning, communicating action plans to patients and following up are operationalised as part of safety netting in English primary care and what approaches are acceptable to both GPs and patients.[46]

### Implications for clinicians and policy-makers
There are several important differences between these three countries, for example, in access to secondary care, in spending on health services and in the numbers of licensed practitioners. The health system in England has less money, fewer doctors and nurses and fewer hospital beds then Sweden or Denmark.[34–36] These factors may yet prove critical in limiting improvements in cancer survival. Patients in Sweden consult their GP less often than patients in England or Denmark but for longer.[36] Shorter consultations may not be efficient—patients in England consult their GPs for nearly 1 hour per year, yet spread over five appointments while, on average, patients in Sweden visit primary care twice a year but for a mean of 20 min per appointment.

Through our international comparison, we have been able to identify opportunities for pathway and service improvement that could promote earlier stage diagnosis. The perception of systemic barriers (eg, access to PCP appointments) may prevent patients seeking timely help, even in the presence of potentially alarming symptoms (eg, blood in stools or a cough that lasts for 3 weeks). While awareness of symptoms may be important, there is a limit to how much effect increasing public awareness could have in settings where access is limited and people are uncertain when to consult or reconsult.[23 26]

The importance of safety netting was emphasised in the 2015 guidance for suspected cancer from the UK National Institute for Health and Care Excellence[50] and a 2011 Delphi study of British GPs and primary care researchers led to publication of suggestions for how to make safety netting more effective.[46] In Denmark, direct access to investigation from general practice, with the aim to lower the GPs' threshold for referring and lengthy 'wait-and-see' approaches,[30] is being investigated along with a focus on safety netting. Our findings suggest that clearly communicated action plans at the end of any primary care consultation could be encouraged, without the need for further research. This would be likely to improve the patient's experience and may also reduce demand for repeated appointments and diagnostic delay.

### Unanswered questions/need for further research

Research is needed into ways of redesigning primary care services in all three countries to reduce (perceived) barriers to consultation. The National Institute for Health and Care Excellence has recommended that British PCPs employ safety netting to reduce delays in the diagnosis of cancer, but there has been little research to understand what constitutes best practice. Further research might address how a 'test of time' or 'wait-and-see' approach that includes clear planning affects the number of follow-up visits and cancer stage at diagnosis. We do not know how action planning is communicated in the consultation nor what effect this may have on the length or frequency of consultations. This could be explored in future mixed-methods research, ideally including conversation analysis of the consultation. Cross-country differences in referral and follow-up practices in patients with lung and bowel cancer could be investigated further, especially in the light of recent studies suggesting that smokers (in Denmark and England) may be less likely to consult. Is the same pattern apparent in Sweden—and if not, why not?

### CONCLUSION

If all primary care consultations concluded with a clear description of what should happen next and the circumstances in which the patient should return, this might avert some of the uncertainty and repeated consultations that were described in these interviews. We conclude that there is an opportunity for clearer communication and better follow-up strategies in primary care—neither of which will put additional pressure on specialist services, yet hold the potential to reduce diagnostic delays with few additional resources.

**Author affiliations**
[1]Health Experiences Research Group, Nuffield Department of Primary Care Health Sciences, University of Oxford, Oxford, UK
[2]The Institute for Palliative Care, Lund University and Region Skåne, Lund, Sweden
[3]Department of Clinical Sciences Lund, Surgery, Lund University, Skane University Hospital, Lund, Sweden
[4]Research Centre for Cancer Diagnosis in Primary Care, Research Unit of General Practice, Aarhus Universitet, Aarhus, Denmark
[5]Department of Learning, Informatics, Management and Ethics, Karolinska Institutet, Stockholm, Sweden
[6]Department of Nursing, Umeå University, Sweden
[7]Department of Infectious Diseases, Karolinska University Hospital, Stockholm, Sweden
[8]School of Health Sciences, City, University of London, UK
[9]Health Services Research Unit, University of Aberdeen
[10]Department of Health Sciences, Lund, Sweden, Lund University, Lund, Skåne, Sweden
[11]Department of Clinical Medicine, University Clinic for Innovative Patient Pathways, Silkeborg Hospital, Aarhus Universitet, Aarhus, Denmark
[12]Center for Innovation, Karolinska Institutet, Stockholm, Sweden

**Acknowledgements** We wish to thank Professor Mike Richards (then National Cancer Director) and Professor Chris Ham (Chief Executive King's Fund) who had the original idea for the study as well as facilitated bringing the three research teams together. We are extremely grateful to the people who took part in this research in all three countries, and to the study advisory panel, including patient and public representatives, who helped design the study and provided comments on an earlier draft of this manuscript. We also acknowledge the support of the National Institute for Health Research, through the Clinical Research Network, who helped recruit patients into the English arm of the study. We would also like to thank all those who helped to recruit participants: In England, we would like to thank the NHS Hospital Trusts that assisted with this study and Patients Active in Research; Thames Valley, along with the following charities who posted links or circulated our details on social media; Beating Bowel Cancer, Bowel Cancer UK, Roy Castle Lung Cancer Foundation, British Lung Foundation and healthtalk.org. In Denmark, we want to thank the Lung Cancer Patient organisation, as well as the Colon Cancer Patient organisation for assisting us in recruiting patients. We also wish to thank the local support groups of the National Danish Cancer Society, as well as the local oncology departments in various regions of Denmark. In Sweden, we would like to thank the nurses and physicians who helped with recruitment.

**Contributors** SZ developed the idea for the manuscript. JM wrote the first draft, with edits made by SZ, JE, MM, CT, RSA and PV. BMB, AC, LE, SH, LL, TON, AC and BR contributed to the analysis meetings and provided comments on drafts.

**Funding** This paper presents independent research funded by organisations from three European countries as follows: In the UK, the study was supported by the National Awareness and Early Diagnosis Initiative (NAEDI). The contributing partners include: Cancer Research UK; Department of Health, England; Economic and Social Research Council; Health and Social Care Research and Development Division, Public Health Agency, Northern Ireland; National Institute for Social Care and Health Research, Wales and the Scottish Government. This funding also covered the costs associated with the comparative analysis meetings in Denmark and Sweden and funded translation of the Danish and Swedish material for publications. During the study LL was Director of Applied Research at the Health Experiences Research Group, Nuffield Department of Primary Care Health Sciences, University of Oxford, and was supported by the National Institute for Health Research (NIHR) Oxford Biomedical Research Centre and the NIHR Collaboration for Leadership in Applied Health Research and Care Oxford (CLAHRC) at Oxford Health NHS Foundation Trust. In Denmark, the study was supported by the Research Centre for Cancer Diagnosis in Primary Care funded by The Danish Cancer Society and the Novo Nordic Foundation. In Sweden, the study was supported by the Vårdal Foundation; the Strategic Research Program in Care Sciences (SFO-V), Umeå University; the Cancer Research Foundation in Northern Sweden and from government funding of clinical research within the National Health Service, Sweden.

**Disclaimer** The views expressed in this paper are those of the authors and not necessarily those of the NAEDI, Danish and Swedish funding partners.

**Competing interests** None declared.

**Patient consent** Obtained.

**Ethics approval** Approvals for research ethics and governance were obtained separately according to the requirements in each country as follows: England: Research Ethics Service reference 14/NS/1035 Denmark: The Biomedical Research Ethics Committee System Act does not apply to this project, as the project does not implicate the use of human biological materials. Standard ethical protocol according to the American Anthropological Association was followed. Sweden: Regional Ethics Board, Lund, Sweden, reg. no 2014/819.

**Provenance and peer review** Not commissioned; externally peer reviewed.

**Data sharing statement** In Denmark the data are available for secondary analysis in conjunction with members of the research group named Comparative Cancer Experiences. In England the participants gave informed consent for data to be copyrighted to the University of Oxford for secondary analysis, broadcasting, publication and teaching. In Sweden the data are available for secondary analysis in conjunction with members of the research group named Comparative Cancer Experiences.

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
