## [Reviewer comments · BMJ Open]

ARTICLE DETAILS

TITLE (PROVISIONAL)	Patients' initial steps to cancer diagnosis in Denmark, England, and Sweden: what can a qualitative, cross country comparison of narrative interviews tell us about potentially modifiable factors?
AUTHORS	MacArtney, John; Malmström, Marlene; Overgaard Nielsen, Trine; Evans, Julie; Bernhardson, Britt-Marie; Hajdarevic, Senada; Chapple, Alison; Eriksson, Lars; Locock, Louise; Rasmussen, Birgit; Vedsted, Peter; Tishelman, Carol; Andersen, Rikke; Ziebland, Sue

VERSION 1 – REVIEW

REVIEWER	Sarah Nettleton University of York, UK
REVIEW RETURNED	29-Jun-2017

GENERAL COMMENTS	This is a clearly presented paper that reports on qualitative study that compares experiences of diagnosis, help seeking and encounters with health care for men and women who have cancer. I have recommended that the paper be accepted and it is a polished piece and so I would be happy to see it accepted as it is. However, I did find the "repeated consultations with patient awareness of the next steps" to be the most novel finding, and as the authors indicate in the subsequent discussion the potentially the most significant for policy and practice. I'm not sure if the paper has reached the word limit - but I did wonder if the authors might extend this subsection a bit to give the readers more feel for how this works in practice. To create words some of the earlier sections could be trimmed - as those findings are consistent with existing research and so while important not quite so novel. This is just a suggestion. Overall a lovely paper which is likely to be impactful.
--

REVIEWER	Dr Nicole Rankin University of Sydney, Sydney Catalyst Translational Cancer Research Centre, Australia Cancer Council NSW, Cancer Research Division, Australia
REVIEW RETURNED	04-Jul-2017

GENERAL COMMENTS	Thank you for the opportunity to review this manuscript. This qualitative interview study provides an analysis of bowel and lung patients' experiences of initial presentation to primary care and identifies potentially modifiable factors within health systems.
---

The strengths of the analysis are evident in the cross-country comparison; the authors highlight similarities in England and Denmark in the lack of specific actions that may prevent people with signs and symptoms of cancer from making timely appointments and the lack of clear actions that follow, in comparison to much clearer planning reported by Swedish participants.

The introduction provides an account of the ICBP studies and a rationale for why further qualitative investigations may explain some of the variation observed in these studies. There is an inconsistent description between objectives in the abstract and those stated in the introduction (page 8 lines 27-34) and these statements should be aligned. The rationale for the selection of bowel and lung cancer is raised in the discussion (“We chose lung and bowel cancer because they affect both genders, are the two most common causes of death from cancer across Europe, their symptomatology is often diffuse or vague, and these cancers are often prone to late stage diagnosis”). This information should be presented in the introduction as it will help inform the reader about the authors’ reasons for conducting the study.

The methods section is thorough in detailing the purposive sampling, analysis and patient involvement. More information about the development of the measure (currently situated in the data collection section) would be helpful. How did the author team develop the semi-structured topic guide across the three research teams? Please clarify whether the team or an individual research developed this initially, and then refined the topics for inclusion through discussion.

The results section provides a descriptive account of the data. Identification of potentially modifiable steps is presented in an eloquent way and appears to provide a balanced account of patient’s experiences across the three countries. Illustrative quotes are appropriately selected to give an impression of the depth of issues covered during the interviews, although more of these are from the bowel cancer patients. The results section does not report on sample characteristics. It appears that little or no demographic information was collected about the participants and while this may be consistent with qualitative sociological research, for the purposes of publishing in a biomedical journal, the readership will anticipate minimal reporting of this information. Please include a table that lists the number of interviews conducted in each for Denmark, Sweden and England, and the number of bowel and lung cancer interviews. It should include data about participant’s sex and age ranges (currently situated in the methods section).

The principal findings presented in the discussion highlight the modifiable steps particularly suggests that in Denmark and England, there is significant opportunity to address the lack of clarity about actions and when to return for consultations. The authors present a thoughtful discussion about this, particularly regarding the use of safety netting. The discussion would be strengthened by signalling how future research efforts might also leverage additional ICBP publications that combine qualitative and quantitative data, or through the development of new interventions.

REVIEWER	Melanie Morris LSHTM, UK
REVIEW RETURNED	04-Jul-2017

GENERAL COMMENTS	Overall This was an interesting and worthwhile study, generally well-written, but needs some revision. General points  • I don't think the title matches the study and findings so well - I'm not sure what is meant by "Initial steps to cancer diagnosis": first presentation at primary care, first attempts at improving diagnosis? - "Variation in experience of cancer diagnosis in primary care in Denmark...etc: what potentially modifiable factors can be identified in qualitative, cross-country comparisons... " • There are some problems with the variation of the sources of recruitment for the participants across countries that are not adequately addressed in the paper, potentially limiting the comparability of the findings across the countries, which therefore need to be justified. The results generally need to be more thoroughly described and the differences and similarities between countries brought out more. Abstract  • For those unfamiliar with the results of the ICBP it is important to describe what patterns the study is hoping to illuminate • PCP – don't use the acronym in the abstract • Results could be summarised more clearly and succinctly (this reflects the need for more clarity in the main Results section) • The Conclusion is overstated as the study has not shown a concrete potential for reduction of these issues, but only highlighted that they exist and hypothesised that they could be reduced with clearer action plans. Article Summary  • Some very long sentences that could be re-written in clearer language • PCP – still has not been explained, don't use the acronym in this summary box Introduction  • p.8 line 14: NAEDI has now been superseded by other initiatives and should not be referred to as if it is current • p.9 line 26: might help to put the year in which Sweden implemented screening as it is given for the others (even if different in different regions, the first date might be informative) Methods  • A breakdown of the characteristics of the people included in the study is needed, either here or, better, at the start of the results: we are not even told how many were interviewed by country. Although we are told there was gender, age, location, pathway "balance", it would have been helpful to see these data summarised, perhaps in a table. • The table could also include source of recruitment: the fact that some participants were recruited from clinics and some from social media, and this was different in the different countries should be shown clearly and discussed as a potential limitation. Patients' deprivation status would also be informative. Were any from hard-to-
---

reach groups?

- Was the stage of each participants' cancer ascertained? This would have had quite a potentially large impact on their responses.

- More detail on the interview protocol would be welcome: how was consistency maintained across interviewers/countries? Was there training done? What "social science theories" were used as the basis of the guide? Perhaps include the guide as an appendix and comment on how closely it was followed in the different settings.

- "theoretical insights" has a large number of references – it would be useful if some of these can be summarised for the reader

- PPI: only English patients looked at the study data. Why none in Denmark or Sweden? If this was not possible, say so and explain.

Results

- The results need more clarity to really bring out the themes and the differences between countries

- Although this is a qualitative study, I was left wondering whether some of the assertions were made by just one or many or most of the participants, in all countries, in two, in one? The themes and variations between countries did not come out convincingly because of this

- One observation given as an illustration of "GP's [should be GPs'] time pressures" really appeared to be a comment about the role of secretaries as gate keepers that need to be navigated (because GPs are busy – but that did not seem to be the point the participant was making)

- The stage of the patients' cancer at diagnosis seemed to me to potentially have an enormous impact on their responses, but it was apparently not considered in the results?

- p17 line 3: How many patients constituted the "planned group" mentioned? How many in the other group? Did this differ between countries? Just in descriptive terms (not for statistical comparison), but we have no idea how many of each there are.

- More explicit comparison between the countries is warranted throughout

Discussion

- p18 line 50: the findings have highlighted potentially modifiable factors, but it is a leap to suggest that they could improve timely diagnosis: a study would need to be done showing that changing the way GPs interact with patients would increase early diagnosis

- p20 line 20-28: this needs more thought – as an epidemiologist I would say it's unnecessary to label recall bias "as epidemiologists would see it". It is either recall bias or it isn't. Explain if it is, leave it out if it isn't.

- if there are two groups: some who saw specialists and some who didn't, then there might be a chance for recall bias – those seeing a specialist might be more motivated to remember things than those who didn't? But you say they all saw a specialist, so this is not what might be happening here?

- I could also see reasons why reflections of experience (not necessarily recall biased) might function differently in the three countries – if the way a specialist interacts with patients differs between countries (part of your hypothesis) then the pleasantness (or otherwise) of the experience itself might influence their likelihood of reporting what happened etc...

	 • Different terms are used in the Discussion for what I think are the same things: “wait-and-see”, “time will tell”. • A limitation that needs discussion is the different source of recruitment for participants: those recruited in clinical settings may well be different from those recruited by social media. Given that the sources were different in different countries (ie there was not the same variety of sources in each country) this could have had a large impact on the results and especially on the comparability between countries. • You mention “avenues of service redesign...which could be implemented without further research”, and similar phrases through the Discussion – what are these? Make some explicit recommendations. Your last sentence says “with few additional resources” but I’m not convinced. “Clearer communication and better follow-up strategies” might not put extra pressure on specialist services but might need training, or at least effective communication of findings to GPs, and probably increased consultation time if GPs are to implement the changes. Perhaps mention what further studies need to be done to take this to the next step.
--	--

REVIEWER	Eila Watson Oxford Brookes University UK
REVIEW RETURNED	14-Jul-2017

GENERAL COMMENTS	I thought this was a well-conducted study, which generated some useful findings. The paper was clearly written, reporting followed accepted criteria for qualitative research, and patient and public involvement was good. I would make the following suggestions for improving the manuscript:  1. The objective of the study was ‘to illuminate patterns observed in the International Cancer Benchmarking Programme Studies in relation to presentation and referral with cancer symptoms’. I felt the relevant findings from the ICBP studies for the three countries included in this study needed to be more clearly stated and also related to the findings presented here. Exactly what patterns have been illuminated? 2. The paper lacks information on the participant characteristics - inclusion of a table indicating for each country, numbers by cancer type, gender, age-range, smoking status, recruitment method etc.would be helpful. Stage at diagnosis would also be helpful information to include if known. The methods state that recruitment was initially done via hospital clinics but supplemented by other recruitment methods. Was the number recruited through hospitals similar in each of the countries? The potential for bias in the sample should be mentioned as a limitation.  3. The backgrounds of the interviewers varied between the countries (anthropologist / sociologist / research nurse) – the paper describes the considerable efforts made to ensure consistency of interviews (and analysis),
--

	but in addition it would be useful to include the semi-structured interview topic guide as an appendix, and also to reflect on the potential for differences in the data between countries arising from differences in interviewing style. 4. In reporting the findings the terms 'more often', 'featured more', 'less prominent' were used to convey observed differences between Sweden and Denmark / England. Whilst appreciating that this is qualitative data and one would not want to quantify, it would be helpful to provide the reader with a better understanding of how you defined 'more'? Are you referring to a large or small difference between the datasets? Were the differences true for both cancer types? 5. In the findings on p12, line 10, the sentence needs editing as it only makes sense for smokers (not 'especially those who smoked'). On p12, line 23 it would be useful to expand further on the self-imposed time limit participants used with regard to symptoms – to what extent were these reasonable time limits? In the section on access, I wondered if there was any sense at all that perceived poor access to primary care in, for example, the UK may be used as an 'excuse' not to attend and to ignore symptoms? How do the authors think this issue of perceived barriers to accessing care can be addressed? 6. Why is the abstract conclusion restricted to safety-netting? Perhaps a space issue, but what about the other factors discussed in the paper?
--	--

VERSION 1 – AUTHOR RESPONSE

Reviewer: 1

Reviewer Name: Sarah Nettleton

Institution and Country: University of York, UK

Please state any competing interests: None

Comment: I did find the "repeated consultations with patient awareness of the next steps" to be the most novel finding, and as the authors indicate in the subsequent discussion the potentially the most significant for policy and practice. I'm not sure if the paper has reached the word limit - but I did wonder if the authors might extend this subsection a bit to give the readers more feel for how this works in practice.

Response

We thank the reviewer for their interest in this novel finding and their encouragement to expand this section. We had reached the word limit with the additions required by the other reviewers, so we looked carefully at this section and concluded that this is not disproportionately brief for this paper. We have also added to the discussion that future research using conversation analysis, could explore action planning in the consultation.

Reviewer: 2

Reviewer Name: Dr Nicole Rankin

Institution and Country: University of Sydney, Sydney Catalyst Translational Cancer Research Centre, Australia

Cancer Council NSW, Cancer Research Division, Australia

Please state any competing interests: None declared

Comment: The introduction provides an account of the ICBP studies and a rationale for why further qualitative investigations may explain some of the variation observed in these studies. There is an inconsistent description between objectives in the abstract and those stated in the introduction (page 8 lines 27-34) and these statements should be aligned.

Response: We agree with the reviewer there is an inconsistency here and have clarified (in italics) that on p7 that the research question we ask is “what might explain some of the variations identified in the ICBP”.

Comment: The rationale for the selection of bowel and lung cancer is raised in the discussion (“We chose lung and bowel cancer because they affect both genders, are the two most common causes of death from cancer across Europe, their symptomatology is often diffuse or vague, and these cancers are often prone to late stage diagnosis”). This information should be presented in the introduction as it will help inform the reader about the authors’ reasons for conducting the study.

Response: We agree with the reviewer that the statement about choice of which cancers to include in the study should be moved and is now in the Introduction (second paragraph).

Comment: More information about the development of the measure (currently situated in the data collection section) would be helpful. How did the author team develop the semi-structured topic guide across the three research teams? Please clarify whether the team or an individual research developed this initially, and then refined the topics for inclusion through discussion.

Response: Although we did not develop a ‘measure’ for this qualitative study we did have regular discussions within the team to agree the semi-structured topic guide. We hope the following addition to the methods helps to clarify how this was done (highlighted here in italics): “Monthly teleconferences with the field research team (all of whom had a high level of spoken and written English) were held throughout the development of the topic guide, recruitment, data collection and analysis phases.” We would also like to add that members of the team have discussed cross-country qualitative methods in more detail in an illustrated literature review – now in press at Qualitative Health Research (Chapple and Ziebland, 2017). A reference to this paper has been provided in the data collection section (p10) and discussion (p20).

Comment: The results section provides a descriptive account of the data. Identification of potentially modifiable steps is presented in an eloquent way and appears to provide a balanced account of patient’s experiences across the three countries. Illustrative quotes are appropriately selected to give an impression of the depth of issues covered during the interviews, although more of these are from the bowel cancer patients.

Response: Of the extended excerpts seven of the twelve extracts concern bowel cancer – these have been carefully selected to illustrate the main analytic points, which did not include contrasting the two cancers.

Comment: The results section does not report on sample characteristics. It appears that little or no demographic information was collected about the participants and while this may be consistent with qualitative sociological research, for the purposes of publishing in a biomedical journal, the readership will anticipate minimal reporting of this information. Please include a table that lists the number of interviews conducted in each for Denmark, Sweden and England, and the number of bowel and lung cancer interviews. It should include data about participant's sex and age ranges (currently situated in the methods section).

Response: We did collect certain socio demographic information and have now included a table of participants' main demographic characteristics (to be inserted on Table 1, page 9; attached as separate file for review.)

Comment: The discussion would be strengthened by signalling how future research efforts might also leverage additional ICBP publications that combine qualitative and quantitative data, or through the development of new interventions.

Response: We thank the reviewer for her encouragement and have added on p22 under Unanswered questions/need for further research, "We do not know how action planning is communicated in the consultation, nor what effect this may have on the length or frequency of consultations. This could be explored in future mixed methods research, ideally including conversation analysis of the consultation". We also hope that our findings will contribute to new policy and practice about the communication of action plans in the primary care consultation.

Reviewer: 3

Reviewer Name: Melanie Morris

Institution and Country: LSHTM, UK

Please state any competing interests: None declared

Comment: I don't think the title matches the study and findings so well - I'm not sure what is meant by "Initial steps to cancer diagnosis": first presentation at primary care, first attempts at improving diagnosis? - "Variation in experience of cancer diagnosis in primary care in Denmark...etc: what potentially modifiable factors can be identified in qualitative, cross-country comparisons... "

Response: We thank the reviewer for this suggestion. After further discussion, we have amended the title to (change in italics): "Patients' initial steps to cancer diagnosis in Denmark, England, and Sweden: what can a qualitative, cross country comparison of narrative interviews tell us about potentially modifiable factors?"

Comment: There are some problems with the variation of the sources of recruitment for the participants across countries that are not adequately addressed in the paper, potentially limiting the comparability of the findings across the countries, which therefore need to be justified.

Response: We agree with the reviewer that further reflection on participant recruitment is needed. We have explained on p9, "To reach data saturation, in England and Denmark this approach was supplemented with some additional recruitment from support groups, social media, and word-of-mouth." Because potential participants were invited to contact the researcher(s) we did not always know how they heard about the study. We have added the following on p19 to Strengths and weaknesses of the study in the Discussion: "Participants who were recruited via social media in Denmark and England may be thought to have had different experiences, than those recruited via clinics. However, while seeking a maximum variation of participants, the focus for this study was that all participants were, or had been, under specialist cancer care and were within six months of diagnosis."

Comment: The results generally need to be more thoroughly described and the differences and similarities between countries brought out more.

Response: We thank the reviewer for raising an issue that was at the centre of the many discussions we had as a research group. A strength of qualitative research is its specificity and in this paper every participant excerpt includes the participant's country. We worked hard as a team to produce an account that balances the need to state clearly the similarities and differences between the three countries, while avoiding overgeneralisation. We believe that we have identified an important difference between consultations in Denmark, England, and Sweden and are keen not to either over reach the data, or muddy the waters, by presenting other comparisons of which we are less confident.

Abstract

- For those unfamiliar with the results of the ICBP it is important to describe what patterns the study is hoping to illuminate
- PCP – don't use the acronym in the abstract
- Results could be summarised more clearly and succinctly (this reflects the need for more clarity in the main Results section)
- The Conclusion is overstated as the study has not shown a concrete potential for reduction of these issues, but only highlighted that they exist and hypothesised that they could be reduced with clearer action plans.

Response: We thank the reviewer for her comments and can note that:

- The objective includes the ICBP patterns concerning "presentation and referral with cancer symptoms" that we sought to illuminate.
- PCP has been removed and replaced with "primary care".
- The conclusion has been amended (changes in italics) to read "We suggest that if clear action plans, as part of safety netting, were routinely used in primary care consultations then uncertainty, false reassurance, and the inefficiency and distress of multiple consultations could be reduced."

Article Summary

- Some very long sentences that could be re-written in clearer language
- PCP – still has not been explained, don't use the acronym in this summary box

Response:

- We agree that the length of the sentences are not ideal, but this section requires single sentences. We have reduced the word count where possible.
- PCP has again been changed to primary care.

Introduction

- p.8 line 14: NAEDI has now been superseded by other initiatives and should not be referred to as if it is current
- p.9 line 26: might help to put the year in which Sweden implemented screening as it is given for the others (even if different in different regions, the first date might be informative)

Response:

- We have removed the reference to NAEDI.
- We have added the following to the paragraph 'The healthcare systems in Denmark, England and Sweden': "In Sweden bowel screening was implemented regionally, with some regions having some form of programme from 2008. Currently it is recommended for 60-74 year olds, although this again varies by region."

Methods

- A breakdown of the characteristics of the people included in the study is needed, either here or, better, at the start of the results: we are not even told how many were interviewed by country. Although we are told there was gender, age, location, pathway “balance”, it would have been helpful to see these data summarised, perhaps in a table.

Response

- A table of participants’ main demographic characteristics has now been included (Table 1, page 9; attached separately for review).

Comment: The table could also include source of recruitment: the fact that some participants were recruited from clinics and some from social media, and this was different in the different countries should be shown clearly and discussed as a potential limitation.

Response: We appreciate why the reviewer considers that we should include these participant details. Unfortunately, we were not able to do so for this study. Potential participants were invited to contact the researcher(s) so we did not always know how they heard about the study. Our aim (as an interview study) was to collect a broad, maximum variation sample in each of the countries. Recruitment was most straightforward in Sweden where the (three) researchers were allowed to approach patients in hospital clinics. In England and Denmark recruitment via clinics was slower. So to reach data saturation (necessary for the cross country comparative study) we used additional approaches including social media (we also address this in Chapple and Ziebland (2017) in press). We have added to our explanation in the methods section on sampling on p8. Through regular teleconferences we were able to recognise and agree when saturation was reached in each country.

Comment: Patients’ deprivation status would also be informative. Were any from hard-to-reach groups? Was the stage of each participants’ cancer ascertained? This would have had quite a potentially large impact on their responses.

Response: We agree with the reviewer that such information could be useful, but we did not collect consistent information about patients’ social class backgrounds (we do know participants’ occupations, but have not routinely included this with extracts). Patients did not always know the stage of their cancer and our study did not seek permission to examine their medical records. We have added the following to the Strengths and weaknesses of the Discussion on p19: “Participants’ cancer disease stage is likely to have affected their experiences. To achieve diversity we recruited people who were diagnosed with very early stage and others diagnosed with advanced disease in all three countries.”

Comment: More detail on the interview protocol would be welcome: how was consistency maintained across interviewers/countries? Was there training done? What “social science theories” were used as the basis of the guide? Perhaps include the guide as an appendix and comment on how closely it was followed in the different settings.

- “theoretical insights” has a large number of references – it would be useful if some of these can be summarised for the reader

Response: We have included the topic guide, in all three languages, as an appendix.

Regarding the development and use of the topic guide, we note on page 9, “The research team had extensive discussions about the topics to ensure comparable data were collected.” and on page 10 (italics added) “Monthly teleconferences with the field research team (all of whom had a high level of spoken and written English) were held throughout the recruitment, data collection and analysis phases.”

The social and health science theories were discussed in the Introduction and so are cross-referenced here. As well as making it clearer in the Introduction which literature we are referring to, we have also amended this sentence to read (changes in italic), “During the interview the researchers used a semi-structured topic guide based on social science theories, and the cancer research literature (highlighted in the Introduction), including factors related to the diagnostic interval.”

Comment: PPI: only English patients looked at the study data. Why none in Denmark or Sweden? If this was not possible, say so and explain.

Response: We note that “Public and Patient Involvement (PPI) was conducted in accordance with good practice in each country” and have added on p11, “In all three countries PPI members were invited to comment on the draft interview topic guide.”

Results

- The results need more clarity to really bring out the themes and the differences between countries
- Although this is a qualitative study, I was left wondering whether some of the assertions were made by just one or many or most of the participants, in all countries, in two, in one? The themes and variations between countries did not come out convincingly because of this
- More explicit comparison between the countries is warranted throughout

Response: As we note above, in this qualitative study our aim is to suggest possible reasons for an observed (by ICBP) difference in cancer survival. During the drafting of this paper we tried various ways of presenting the findings and have concluded that the current description appropriately reflects the findings without over stating other (more marginal) differences between the countries. Our approach is in line with the nature of the data and the aims of the study and reflects the experience of the team – we sympathise with the reviewer’s interest in more concrete comparisons but have suggested that different research methods, including mixed methods, would be needed to address these.

Comment: One observation given as an illustration of “GP’s [should be GPs] time pressures” really appeared to be a comment about the role of secretaries as gate keepers that need to be navigated (because GPs are busy – but that did not seem to be the point the participant was making)

Response: We agree that this could also be interpreted as being about both reception staff behaviour and GP time pressures. What we believe we have shown here is how the participant relates time constraints to practice culture and reception behaviour. The full quote is, “The system is not always that easy. First you have to convince the secretary, that you need an appointment, right. That is what happens when they are too busy” (italics added).

Comment: The stage of the patients’ cancer at diagnosis seemed to me to potentially have an enormous impact on their responses, but it was apparently not considered in the results?

Response: As we note above, we agree with the reviewer that such information may be informative. However, the information we collected about patients’ disease stage was not sufficiently consistent to include. Patients did not always know the stage of their cancer and our study did not have permission to examine their records. To reflect this, we have added the following to the Strengths and weaknesses of the Discussion: “Participants’ cancer disease stage may have affected their experiences. For this reason, we recruited people who were diagnosed with very early stage and others diagnosed with advanced disease.”

Comment: p17 line 3: How many patients constituted the “planned group” mentioned? How many in the other group? Did this differ between countries? Just in descriptive terms (not for statistical comparison), but we have no idea how many of each there are.

Response: During the drafting of this paper we tried various ways of presenting the findings (including various frequency based formats) and have concluded that the current description most appropriately reflects the findings without over stating other (more marginal) differences between the countries. Our approach is in line with the nature of maximum variation sample, the data and the aims of the study, and reflects the experience of the team. We share the reviewer’s interest in numerical comparisons, but concluded that different research methods would be needed to address these.

Comment: p18 line 50: the findings have highlighted potentially modifiable factors, but it is a leap to suggest that they could improve timely diagnosis: a study would need to be done showing that changing the way GPs interact with patients would increase early diagnosis.

Response: We agree that we would be overstating our findings to suggest that we have identified the solution to cross country differences in cancer survival. However, having discussed the findings and the paper across the whole research group, it is our opinion that we have been very careful to avoid this. Rather we suggest that clear communication of an action plan at the end of every GP consultation (whatever the condition) is unlikely to have adverse effects.

Comment: p20 line 20-28: this needs more thought – as an epidemiologist I would say it’s unnecessary to label recall bias “as epidemiologists would see it”. It is either recall bias or it isn’t. Explain if it is, leave it out if it isn’t. - if there are two groups: some who saw specialists and some who didn’t, then there might be a chance for recall bias – those seeing a specialist might be more motivated to remember things than those who didn’t? But you say they all saw a specialist, so this is not what might be happening here? I could also see reasons why reflections of experience (not necessarily recall biased) might function differently in the three countries – if the way a specialist interacts with patients differs between countries (part of your hypothesis) then the pleasantness (or otherwise) of the experience itself might influence their likelihood of reporting what happened etc...

Response: We are agreed that reference to recall bias is unnecessary here, given the nature of the study. We have changed the sentence to: “All participants were, or had been, under specialist cancer care, which may have affected the way they reflected upon the time and events that passed before their diagnosis, although we sought a range of experiences of specialist care across the three countries.”

Comment: Different terms are used in the Discussion for what I think are the same things: “wait-and-see”, “time will tell”.

Response: We thank the reviewer for highlighting this. We note that we also used “test of time” (p17) in the findings, (citing Almond et al, 2009). We can confirm that this reflects different common usage in the three countries (for example, in Denmark “wait-and-see” is the best translation for the practice). To ensure consistency and clarity we have removed “time will tell”, and now use both “test of time” and “wait-and-see” (with appropriate references) in the Findings (p17) and Discussion (p22).

Comment: A limitation that needs discussion is the different source of recruitment for participants: those recruited in clinical settings may well be different from those recruited by social media. Given that the sources were different in different countries (ie there was not the same variety of sources in each country) this could have had a large impact on the results and especially on the comparability between countries.

Response: We agree with the reviewer that further reflection on participant recruitment is needed. We note that potential participants were invited to contact the researcher(s) so we did not always know how they heard about the study. However, we have added the following to Strengths and weaknesses of the study in the Discussion: "Participants who were recruited via social media in Denmark and England may also be thought to have had different experiences, than those recruited via clinics. All participants were, or had been, under specialist cancer care and within six months of diagnosis."

Comment: You mention "avenues of service redesign...which could be implemented without further research", and similar phrases through the Discussion – what are these? Make some explicit recommendations. Your last sentence says "with few additional resources" but I'm not convinced. "Clearer communication and better follow-up strategies" might not put extra pressure on specialist services but might need training, or at least effective communication of findings to GPs, and probably increased consultation time if GPs are to implement the changes. Perhaps mention what further studies need to be done to take this to the next step.

Response: We have amended the sentence to include the following (in italics): "While the high level comparisons of large quantitative data bases and surveys provide invaluable information, qualitative studies can suggest why these patterns may occur and identify avenues for service redesign, some of which such as routine GP action plans, could be implemented without further research."

Reviewer: 4

Reviewer Name: Eila Watson

Institution and Country: Oxford Brookes University, UK

Please state any competing interests: None declared

1. The objective of the study was 'to illuminate patterns observed in the International Cancer Benchmarking Programme Studies in relation to presentation and referral with cancer symptoms'. I felt the relevant findings from the ICBP studies for the three countries included in this study needed to be more clearly stated and also related to the findings presented here. Exactly what patterns have been illuminated?

Response: We have amended the paper to make it clear where we are discussing the potentially modifiable factors, as highlighted in some ICBP findings. For example, in the Introduction we now note (changes in italics), "These studies have shown a number of potentially modifiable factors, for example, that patterns in public knowledge about cancer awareness and beliefs were not clearly associated with variations in survival across countries.¹"

2. The paper lacks information on the participant characteristics - inclusion of a table indicating for each country, numbers by cancer type, gender, age-range, smoking status, recruitment method etc. would be helpful. Stage at diagnosis would also be helpful information to include if known.

Response: A table of the participants' demographic characteristics that were available to us has now been included (Table 1, page 9). Patients did not always know the stage of their cancer and our study did not seek permission to examine their medical records. We have added the following to the Strengths and weaknesses of the Discussion on p19: "Participants' cancer disease stage is likely to have affected their experiences. To achieve diversity we recruited people who were diagnosed with very early stage and others diagnosed with advanced disease in all three countries."

Comment: The methods state that recruitment was initially done via hospital clinics but supplemented by other recruitment methods. Was the number recruited through hospitals similar in each of the countries? The potential for bias in the sample should be mentioned as a limitation.

Response: As we note in our responses above, repeated here for ease, we agree with the reviewer that further reflection on participant recruitment is needed. We have explained on p9, “To reach data saturation, in England and Denmark this approach was supplemented with some additional recruitment from support groups, social media, and word-of-mouth.” Because potential participants were invited to contact the researcher(s) we did not always know how they heard about the study. We have added the following on p19 to Strengths and weaknesses of the study in the Discussion: “Participants who were recruited via social media in Denmark and England may be thought to have had different experiences, than those recruited via clinics. However, while seeking a maximum variation of participants, the focus for this study was that all participants were, or had been, under specialist cancer care and were within six months of diagnosis.”

3. The backgrounds of the interviewers varied between the countries (anthropologist / sociologist / research nurse) – the paper describes the considerable efforts made to ensure consistency of interviews (and analysis), but in addition it would be useful to include the semi-structured interview topic guide as an appendix, and also to reflect on the potential for differences in the data between countries arising from differences in interviewing style.

Response: We have included the topic guide, in all three languages, as an appendix. We have also added the following to Strengths and weaknesses of the study in the Discussion: “Along with the teleconferences, these meetings helped to mitigate any potential for differences in the data arising from differences in country and disciplinary approaches to research, such as interviewing styles”.

4. In reporting the findings the terms ‘more often’, ‘featured more’, ‘less prominent’ were used to convey observed differences between Sweden and Denmark / England. Whilst appreciating that this is qualitative data and one would not want to quantify, it would be helpful to provide the reader with a better understanding of how you defined ‘more’? Are you referring to a large or small difference between the datasets? Were the differences true for both cancer types?

Response: As the reviewer appreciates, the language used to express degrees of difference in qualitative research can be contentious. These terms are relative and need to be contextualised within the data, which we believe the excerpts used to support our careful presentation of the findings help to do. While there are certainly differences between the experiences of the two cancers, these were not apparent for this particular focus upon modifiable factors at presentation and referral in primary care. We think this is an interesting strength of the study (which might suggest transferability of the findings to other cancers).

5. In the findings on p12, line 10, the sentence needs editing as it only makes sense for smokers (not ‘especially those who smoked’).

Response: We thank the reviewer for highlighting this. It has now been amended.

Comment: On p12, line 23 it would be useful to expand further on the self-imposed time limit participants used with regard to symptoms – to what extent were these reasonable time limits?

Response: We agree with the reviewer that this could be an interesting avenue to explore however regrettably this is not a question we can answer with our dataset.

Comment: In the section on access, I wondered if there was any sense at all that perceived poor access to primary care in, for example, the UK may be used as an ‘excuse’ not to attend and to ignore symptoms? How do the authors think this issue of perceived barriers to accessing care can be addressed?

Response: We agree with the reviewer that this does appear to be an issue for some, as we explore on p12 and cross-reference with ICBP study, Forbes et al., (2013). We have added a further relevant reference (Llanwarne et al., 2017 'Wasting the doctor's time? A video-elicitation study with patients in primary care' Social Science and Medicine) to the discussion.

6. Why is the abstract conclusion restricted to safety-netting? Perhaps a space issue, but what about the other factors discussed in the paper?

Response: While the word count is a factor, we have also focussed the conclusion in the abstract on action plans, which we are keen to highlight as a modifiable change to practice, and (we believe) are the most original contribution from the study.

VERSION 2 – REVIEW

REVIEWER	Nicole Rankin University of Sydney, Sydney Catalyst Translational Cancer Research Centre, Australia Cancer Council NSW, Cancer Research Division, Australia
REVIEW RETURNED	23-Aug-2017

GENERAL COMMENTS	Thank you for opportunity to review the revised manuscript. I have carefully checked the responses to all reviewers' comments and consider that all relevant issues have been thoroughly considered and changes have been made to the manuscript, with one exception. Table 1 should appear in the results and not the methods section (as previously advised). At the time of planning the study, the final number of participants was not known and therefore, should not be a methodological feature of the article. With this minor change, the manuscript should be accepted for publication.
--

REVIEWER	Melanie Morris London School of Hygiene and Tropical Medicine, UK
REVIEW RETURNED	15-Aug-2017

GENERAL COMMENTS	I appreciate the fact that the authors have given full and clear responses to all the comments made - thank you. The paper, which was already of a high standard, is certainly improved. I still feel that there are areas that could have been expressed more clearly, especially in the results, and some limitations inherent in the study design (which were perhaps unavoidable). But these have mainly been discussed and any real confusions or omissions have been dealt with. I do think Table 1 is helpful, but needs proportions as well as numbers for easy, one-glance comparisons between groups. Having read the authors' explanations in their responses, and having now read the paper carefully several times, I feel I understand the study and the conclusions drawn. I do think readers will still have some of the questions I originally had, but without the benefit of the responses to read.
---

	For instance, it is not just that I am "interested in numerical comparisons" (although I am!) but I wanted to feel more of the weight of the evidence behind the assertions made. I will leave it to the editors now to decide whether the themes come across clearly enough for their readership.
--	--

REVIEWER	Eila Watson Oxford Brookes University, UK
REVIEW RETURNED	01-Sep-2017

GENERAL COMMENTS	Reviewer comments satisfactorily addressed.
---

VERSION 2 – AUTHOR RESPONSE

Reviewer: 2

Nicole Rankin

Comment: "Table 1 should appear in the results and not the methods section."

Response: Table 1 has been moved to the results section.

Reviewer: 3

Melanie Morris

Comment: "I still feel that there are areas that could have been expressed more clearly, especially in the results, and some limitations inherent in the study design (which were perhaps unavoidable). But these have mainly been discussed and any real confusions or omissions have been dealt with. I do think Table 1 is helpful, but needs proportions as well as numbers for easy, one-glance comparisons between groups.

"Having read the authors' explanations in their responses, and having now read the paper carefully several times, I feel I understand the study and the conclusions drawn. I do think readers will still have some of the questions I originally had, but without the benefit of the responses to read. For instance, it is not just that I am "interested in numerical comparisons" (although I am!) but I wanted to feel more of the weight of the evidence behind the assertions made. I will leave it to the editors now to decide whether the themes come across clearly enough for their readership."

Response:

We have amended the table to provide proportions to aid with comparison as suggested.

We thank the reviewer for her careful consideration of our paper and that she is satisfied that her prior concerns with the results and limitations of the study have been suitably addressed. We have sought to respond to the helpful comments of all of the reviewers in accordance with the methodological limits of the study. We hope the editor agrees that in doing so we have maintained a balance of presenting our study clearly, while not overstating our findings.